# Determination of Magnetic Structures in Magnetic Microwires with Longitudinally Distributed Magnetic Anisotropy

**DOI:** 10.3390/s23063079

**Published:** 2023-03-13

**Authors:** Alexander Chizhik, Paula Corte-Leon, Valentina Zhukova, Julian Gonzalez, Przemyslaw Gawronski, Juan Mari Blanco, Arcady Zhukov

**Affiliations:** 1Department Advanced Polymers and Materials: Physics, Chemistry and Technology, University of Basque Country, UPV/EHU, 20018 San Sebastian, Spain; 2Department of Applied Physics, University of Basque Country EIG, UPV/EHU, 20018 San Sebastian, Spain; 3Faculty of Physics and Applied Computer Science, AGH University of Science and Technology, 30-059 Cracow, Poland; 4IKERBASQUE, Basque Foundation for Science, 48011 Bilbao, Spain

**Keywords:** soft magnetic materials, amorphous magnetic microwires, magnetic domains, magneto-optic Kerr effect, magnetic anisotropy

## Abstract

We studied the magnetic properties of a glass-covered amorphous microwire that was stress-annealed at temperatures distributed along the microwire length. The Sixtus-Tonks, Kerr effect microscopy and magnetic impedance techniques have been applied. There was a transformation of the magnetic structure across the zones subjected to annealing at different temperatures. The annealing temperature distribution induces the graded magnetic anisotropy in the studied sample. The variety of the surface domain structures depending on the longitudinal location has been discovered. Spiral, circular, curved, elliptic and longitudinal domain structures coexist and replace each other in the process of magnetization reversal. The analysis of the obtained results was carried out based on the calculations of the magnetic structure, assuming the distribution of internal stresses.

## 1. Introduction

The Giant Magnetoimpedance (GMI) effect is determined by the remarkable change in the magnetic impedance in a wide range of frequencies when an external magnetic field is applied. This allows its sustainable and active use in magnetic, chemical and biochemical sensors. Magnetic wires, in which this effect is observed, have certain advantages that allow them to occupy a special place in the line of active elements [1,2,3,4,5].

Another application of microwires, which we took into account when choosing topics for this article, was their possible use in magnetic memory structures. The idea of controlled motion of domain walls is promising precisely for nano- and microwires, in which the observation of the formation and motion of domain walls is widely studied by a variety of experimental methods [6].

As regards the background of the current research, a new direction of research related to the creation of microwire samples with longitudinally distributed magnetic properties should be noted here. As shown earlier, a characteristic feature of amorphous materials (and particularly of microwires) is that the anisotropy caused by stress annealing depends on the key parameter of the annealing temperature [7,8,9]. The dependence of the magnetic anisotropy caused by stress annealing in microwires on the annealing temperature was proposed for the development of gradient magnetic anisotropy [9,10].

The mentioned distributed magnetic anisotropy is achieved by a fairly simple method, which consists of annealing at a fixed applied mechanical stress and a variable annealing temperature. It is also important that such stress annealing is only partially reversible: subsequent annealing can restore only part of the anisotropy caused by stress annealing [9,10]. In addition, the induced magnetic anisotropy is proportional to the annealing temperature. Accordingly, the parameter that is easiest to modify during stress annealing is the annealing temperature. Therefore, to create microwire samples with smooth magnetic anisotropy, stress annealing in the annealing temperature gradient was proposed [9,10].

In this article, we present our recent results on the study of the magnetic structure and magnetization reversal processes in magnetic microwires with distributed magnetic anisotropy caused in turn by distributed annealing temperature. The originality and novelty of this research is initially determined by our original idea to create samples with a continuous and uniform distribution of magnetic properties. The longitudinal distribution of the annealing temperature is a natural reason for the distribution of magnetic anisotropy, which in turn leads to a continuous distribution of magnetic properties along the length of the microwire. It is the continuity of changes in properties that is the object of our close scientific interest. Having carried out the first studies of such samples, in particular, the influence of graded magnetic anisotropy on domain wall propagation, we discovered the lack of knowledge about the features of the magnetic structure in this type of sample. Although we previously studied the magnetic structure in various annealed samples, it is the continuity of the change in the magnetic properties of the microwire in a wide temperature range from room temperature to 300 °C that is the decisive factor. Previously, we “locally” examined various annealed samples, and, as it turned out, thereby missed and did not fix various essential properties. Now this shortcoming has been eliminated. In particular, for the first time we observe the effect of the coexistence of various magnetic structures, replacing each other both in the process of magnetization reversal and when moving along the sample.

## 2. Experimental Details

We studied a glass-coated microwire with a chemical composition of Co_64.04_Fe_5.71_B_15.88_Si_10.94_Cr_3.4_Ni_0.3_ (diameter of metallic nucleus d = 94 μm, total diameter with glass covering D = 126 μm) prepared by the Taylor-Ulitovsky technique (Figure 1). The long sample was manufactured to ensure a uniform distribution of magnetic properties along the microwire length. The homogeneous microwire was subjected to annealing at variable temperature to obtain the microwire sample with magnetic properties distributed along the microwire length.

During the annealing process, a part of a sample was placed in a conventional furnace, while the rest of the sample remained outside the furnace. A tensile stress was applied during annealing. As a result, different parts of the sample were subjected to annealing at different temperatures (Figure 2). A tensile stress (*σ* = 400 MPa) was created by attaching a mechanical load to one end of the microwire. The temperature, *T_ann_*, inside the furnace was set to 300 °C. The annealing was carried out in air because insulating and continuous glass coatings protect microwires against oxidation. 

The temperature was measured using a commercial (NiCr-Ni KEYSIGHT Technologies, Santa Rosa, CA, USA) thermocouple inside the furnace, in the orifice (through which the microwire is introduced into the furnace) and near the furnace. As a result of the performed measurement, the distribution of the annealing temperature was obtained (Figure 2). Longitudinal homogeneity of the chemical composition and the geometric dimensions allow us to count on a smooth temperature change between room temperature and 300 °C over a sufficiently long length of the wire under study. The additional validity of the obtained temperature dependence was confirmed by the fact that in the course of the study we found a large variety of magnetic structures that smoothly replaced one another.

The magneto-optical technique (Kerr effect) has been applied as a method to retrieve the information about the magnetization reversal (surface hysteresis loops) and to acquire the contrast images of the surface magnetic domain structures [11]. Appling the magnetic field (H_AX_) co-directed with the microwire axis we have observed and fixed by microscope the surface magnetic structure transformation.

To measure the velocity of the volume domain wall we applied the modified method of Sixtus-Tonks [10,12]. The domain wall motion induces the electromotive force (EMF) peaks in three secondary coils. The rectangular pulses of the magnetic field induced the domain wall displacement. The pulses of the magnetic field were produced in turn by the long solenoid. The velocity was determined by the time distance between the (EMF) peaks.

Furthermore, after obtaining the information of the shape of the EMF peaks for different locations in the long microwire sample, we use the Sixtus-Tonks technique for the estimation of the structure of the domain wall (DW) corresponding to the different locations in the studied microwire.

The impedance dependencies were evaluated using the specially designed micro-strip sample holder. The holder was placed inside a sufficiently long solenoid that creates a homogeneous magnetic field. The microwire impedance was determined from the reflection coefficient measured by the vector network analyzer [13,14]. We evaluated the magnetic field dependencies of the GMI ratio, ΔZ/Z, defined as:ΔZ/Z = [Z(H) − Z(H_max_)]/Z(H_max_)(1)
where H_max_ is the maximum applied magnetic field. Use of the mentioned technique allows us to measure the GMI effect in the extended frequency range up to 1 GHz.

The GMI dependencies were measured for the locations in the studied microwire corresponding to the room temperature and to the annealing temperature of 300 °C. The small samples for GMI measurements were cut off from two opposite ends of the long sample, where the regions of interest to us were located. Other parts of the sample were not destroyed during this manipulation.

The crystallization and Curie temperatures of the studied microwire, evaluated by the differential scanning calorimetry (DSC) method as described elsewhere [15], were about 510 and 370 °C, respectively. The crystallization and Curie temperature values are consistent with the values observed in CoFe microwires [16] and ribbons [17] with similar compositions. Consequently, like other Co-rich stress-annealed microwires, the studied microwires annealed at 300 °C retain an amorphous structure after stress annealing [18]. The amorphous structure of stress-annealed microwires is also indirectly confirmed by their excellent magnetic softness. In the X-ray diffraction (XRD) pattern of the studied sample, a wide halo typical for amorphous materials can be observed.

## 3. Results and Discussion

Figure 3 presents the EMF peak transformation obtained from one of the pick-up coils when the sample moves inside the measuring system. In other words, as is shown in Figure 3, the transformation of the EMF peak with the annealing temperature is observed. Specifically, we have observed that starting from room temperature, as the annealing temperature increases, the shape of the peak changes significantly: the peak width increases, and the peak amplitude decreases until the peak disappears. In general terms, this means that the velocity associated with the passage of the DW along the pick-up coil decreases, while the width of the DW probably increases.

Thus, the preliminary results obtained using the Sixtus-Tonks method were the reason for the using of the magneto-optic Kerr effect (MOKE) method for a more detailed study of the magnetization reversal and the rearrangement of the domain structure. The difference between these two methods is that the Sixtus-Tonks method provides information about the passage of a bulk domain wall along the whole cross section of the microwire, while the MOKE method shows the details of the domain structure and the magnetization reversal process in the microwire surface.

Figure 4 demonstrates the local MOKE hysteresis loops obtained in different places on the surface of the long sample annealed at different temperatures. Moving along the sample from room temperature upwards, we found a noticeable transformation of the MOKE hysteresis loop. As we will demonstrate below, there is a certain correlation between the results presented in Figure 3 and Figure 4. Although this correlation is not obvious at first glance, we will demonstrate it during the comparative analysis of the obtained results.

A certain asymmetry of the hysteresis curves is observed at room temperature (RT) and 40 °C, while at temperatures of 60, 140 and 240 °C they are symmetrical enough. Being rectangular or nearly rectangular at temperatures close to room temperature, the MOKE hysteresis curves become progressively flatter as we move along the long sample towards the locations corresponding to higher temperatures of the annealing.

A decisive addition, without which a detailed analysis could not be carried out, is that of the MOKE magnetic images of the structures obtained in the same places of the microwire where the MOKE hysteresis curves were measured (Figure 5). The mentioned asymmetry of the MOKE curves demonstrated in Figure 4 manifests itself in the difference in magnetic domain structures observed during the magnetization reversal that occurred when the external magnetic field was changed from + HZ to − HZ and vice versa. Let us consider successively in detail the variety of the domain structures and the transformation observed on the surface of the microwire as we move along the sample.

(a)Room temperature. Here we have determined that image 4b corresponds to the sharp jump in the MOKE hysteresis (right semi-loop of Figure 3a), while image 4a corresponds to the relatively slow magnetization reversal in the same MOKE hysteresis curve (left semi-loop of Figure 4a).

Figure 5b is the evidence that the magnetization reversal occurs as the fast motion of the inclined elliptic domain wall. Figure 5a reflects a relatively slow formation and the transformation of the multi-domain structure.

(b)Temperature of 40 °C. At this temperature, we also observe two different domain structures corresponding to two semi-loops (Figure 4b). The very sharp jump (right semi-loop of Figure 4b) is associated with quick motion of the longitudinally oriented DW (Figure 5d). On the reverse side of the MOKE loop, the motion of the long curved, almost linear DW is observed (Figure 5c). This DW we qualified as the variation of the long spiral DW observed earlier.(c)For the location on the surface of the sample corresponding to the annealing temperature of 60 °C, the MOKE hysteresis loop is symmetrical (Figure 4c). This means that the magnetization reversal occurred in similar ways when the external magnetic field was reversed. Figure 5e,f correspond to this process of the magnetization reversal. This process consists of two stages. At the first stage, the low formation and transformation of the multi-domain structure takes place (Figure 5e). At the second stage (sharp jump in Figure 4c), the rapid movement of the curved DW occurs (Figure 5f). Visually, this is observed in such a way that the curved DW “washes away” the slow multi-domain structure formed earlier at the first stage of the magnetization reversal.(d)The magnetization reversal process in the location corresponding to the annealing temperature of 140 °C is also a combination of two types of domain structure (Figure 5g). We observed two processes that occurred simultaneously, or succeeded one another. The first process is, as in the case of the 60 °C, the formation and slow transformation of the multi-domain structure placed almost perpendicular to the microwire axis. The second process is the quick jump-like motion of the circular DW moving along the microwire surface. The presence of this type of jump is probably associated with the presence of defects on the surface of the microwire adjacent to the glass coating.(e)Finally, in the location on the surface of the microwire corresponding to the annealing temperature of 240 °C, we found the domain structure which could be tentatively named “slow spiral structure,” which is characterized namely by the relatively low velocity of the motion along the microwire surface. Figure 5h–j demonstrate the consistent motion of slightly inclined spiral domains. The red dashed line shows the position of the front separating the mono-domain part from the part filled by the spiral structure. The arrow shows the direction of movement of this front. This slow movement corresponds to a smooth curve without jumps (Figure 4e).

Additional information for the analysis of the magnetic structures in different locations of the sample is provided by a cross-comparison of Figure 3, Figure 4 and Figure 5. Obviously, the sharp EMF peak presented in Figure 3a corresponds to the fast motion of the inclined elliptic DW. It should be noted here that we were able to measure by the Sixtus-Tonks technique the velocity of the domain wall motion only in the zone corresponding to room temperature (Figure 6). It was possible to carry this out only due to the fact that in this area, there was a movement of the inclined, compact and narrow domain wall.

The wider and lower but still compact EMF peak (Figure 3b) corresponds to the rapid motion of the curved domain wall shown in the Figure 4c. Because of its varying curvature, it has a finite length, unlike the “infinite” longitudinal domain wall shown in Figure 5d. Therefore, a compact EMF peak corresponding to the fast motion of the domain wall is observed in this location. The rectangular shape of the MOKE hysteresis loop (Figure 4b) is in agreement with this consideration. The sharp jumps of the surface magnetization correspond to the fast motion of the longitudinally oriented or curved DWs.

We attribute the appearance of a double EMF peak observed at temperatures 60 °C and 140 °C to the coexistence of two types of domain structure in the process of magnetization reversal. The overall decrease in the amplitude of the EMF peak is associated with the turn of the magnetization in the magnetic domains in the direction of the circular direction and a decrease in the axial projection of the magnetization. For the temperature of 60 °C, one peak is associated with the formation of a multi-domain structure, and the second peak is associated with the rapid motion of a curved domain wall. For the temperature of 140 °C, the first peak is also related to the formation of the multi-domain structure, and the second peak reflects the motion of the circular domain wall separating the multi-domain and mono-domain magnetic state.

Finally, for the location on the microwire surface corresponding to the annealing temperature of 240 °C, the results presented in Figure 3, Figure 4 and Figure 5 are consistent. In this case, we are dealing with a magnetization reversal process determined mainly by the slow movement of the helical magnetic structure along the surface of the microwire. First, the very small amplitude of the EMF peak (Figure 3e) is determined by the fact that the magnetization inside the helical domains is rather close to the circular direction. Secondly, this also determines the low magnetic contrast, since the longitudinal MOKE is sensitive precisely to the axial magnetization. In addition, the slow smooth motion of the spiral domain structure over the surface of the microwire determines the smooth, jump-free shape of the MOKE hysteresis (Figure 4e).

Analyzing the obtained results, we identified several essential factors. For all the studied longitudinal locations on the surface of the microwire, the magnetization reversal process includes the simultaneous or successive coexistence of different types of domain magnetic structures. These structures can be divided into two main classes: those that change rapidly and those that change slowly. Behind this formal difference lies a more significant difference in structure, which we will discuss further.

To explain the results obtained, we involved the results of theoretical simulations [19], as well as the results of MOKE microscopy of the microwire sample polished to the shape of a semi-cylinder.

Experimental MOKE images presented in Figure 7 demonstrate the axial magnetic field transformation of the domain structure on the plane surface of the half-tube shaped microwire. Schematic insets help to recognize this transformation. Depending on the value of the axial magnetic field, there are two types of domain structures located near the cylindrical surface of the half-tube. In Figure 7a these domains have direct correlation with the central part of the microwire. A change in the magnetic field (Figure 7b) leads to the formation of another type of domain—an isolated domain not connected with the central part of the microwire (shown by the red dotted line in the figure).

Theoretical calculations confirm this consideration. Figure 8 shows the calculated domain structure on the flat surface of the semi-cylinder and on the adjacent cylindrical surface. It should be noted that initially the calculations were carried out for all points inside the whole cylinder, and the results are shown in the form of a semi-cylinder for clarity of the analysis. Red and blue colors correspond to two opposite axial directions of the magnetization. As in the case of the experimentally observed domain structure in the half-cylinder (Figure 7), here we also observe two types of domains: red domains correlated with the internal part (modified inner core) and blue domains, independent from the inner domains.

Applying the performed analysis to the experimental results presented in Figure 4, we draw the following conclusions about the nature of the observed surface domain structures. We believe that the surface domain structures associated with the internal domains tend to change slowly (we called them “slow domains”). At the same time, domains located only on the microwire surface have a higher mobility due to their locality, isolation and simpler shape (we called them “quick domains”).

In the location of the microwire determined by the room temperature annealing, we find fast and slow elliptical structures. In the location where the annealing temperature was 40 °C, we find fast longitudinal and curved structures, which are special cases of the spiral domain structure. At an annealing temperature of 60 °C, one structure is replaced by another (the slow elliptical structure is replaced by the fast spiral curved structure) during one half-cycle of magnetization reversal.

The area of the microwire corresponding to the annealing temperature of 140 °C is characterized by the slow formation of an elliptical domain structure close to a circular one and the rapid movement of a single circular domain wall, which apparently has no correlation with the internal structure of the microwire.

Finally, the domain structure observed in the longitudinal location associated with the annealing temperature of 240 °C is the spiral domain structure, with a small angle of inclination of the magnetization from the circular direction. Here we observe, namely, the motion of the whole domain structure, but not the domain wall. The domain structure moves by a formed front (marked by the red dashed line). According to our consideration, the low mobility of the motion of the front, separating the spiral domain structure and the mono-domain state, is determined by the strong correlation of this structure with the modified inner core inside the microwire.

Taking into account the active use of microwires in magnetic sensors and for the sake of completeness of the analysis, we presented the results of GMI measurements carried out at two limit locations of the microwire. These locations correspond to room temperature and the annealing temperature of 300 °C (Figure 9). It is significant that the GMI effect is observed in these two locations. Naturally differing somewhat in the ΔZ/Z-value, these two areas, however, give fairly high GMI ratios. This allows us to consider this microwire to be a promising candidate as an active element for magnetic sensors using the GMI effect.

The difference in the value of the GMI ratio in the two studied locations is explained within the framework of the previously established relationship between the value of the GMI ratio and the angle of the helicity of the surface magnetic structure [20]. As shown in [20], in the region of helical angles of 10−60 °C, there is a direct dependence of the GMI value on the value of the helical angle. As we can see, the angle of the helical structure at RT is greater than the angle of helicity at 300 °C. This difference finds its manifestation in the observed difference of GMI.

Along with sensors based on the GMI effect, microwires are active elements for sensors that use the magnetization reversal that occurs due to the fast motion of domain walls. In this regard, we would like to recall that, in addition to a significant GMI effect, a fast movement of a solitary domain wall is also observed in the location corresponding to RT. This rather unique property makes this area of the microwire the object of our further scientific attention in the frame of the sensor application of magnetic microwires.

## 4. Conclusions

Our extensive research was aimed at the study of the magnetic properties of microwires with distributed magnetic properties. These distributed properties were determined by the distribution of magnetic anisotropy, which in turn was determined by the process of annealing at variable temperature. In our opinion, the most significant result obtained is that we managed to recognize a wide variety of the complex magnetic structures in various longitudinal locations on the surface of the microwire.

We succeeded in determining the basic differences between these magnetic structures, which are manifested in the features of the magnetization reversal process. The complex use of the techniques allowed us to look at the problem from an unexpected angle and find those basic features of magnetic structures that were implemented in all the applied experimental techniques.

To the analysis of the obtained results, we involved the results of well-proven theoretical calculations and experimental images of domain structures observed in magnetic semi-cylinders. As we were able to establish, the investigated domain structures can be divided into two classes, which are characterized by the velocity of their transformation and motion. We determined that the reason for this is the different degree of coupling of surface domains with the internal domain structure. The combination of helical, elliptical, curved, axial and circular domain structures changes smoothly as the point of observation moves along the microwire surface from room temperature to a maximum annealing temperature of 300 °C.

Additionally, the studies of the domain wall velocity and GMI response give us the additional grounds to consider the studied microwire as a possible active element of magnetic sensors.

## Figures and Tables

**Figure 1 sensors-23-03079-f001:**
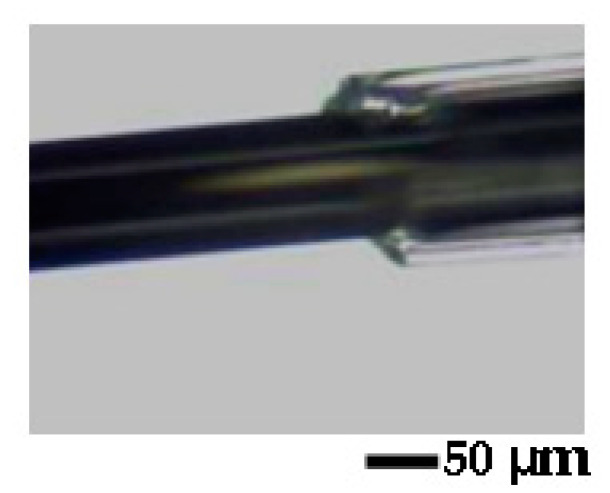
Images of studied sample. The glass coating has been removed intentionally.

**Figure 2 sensors-23-03079-f002:**
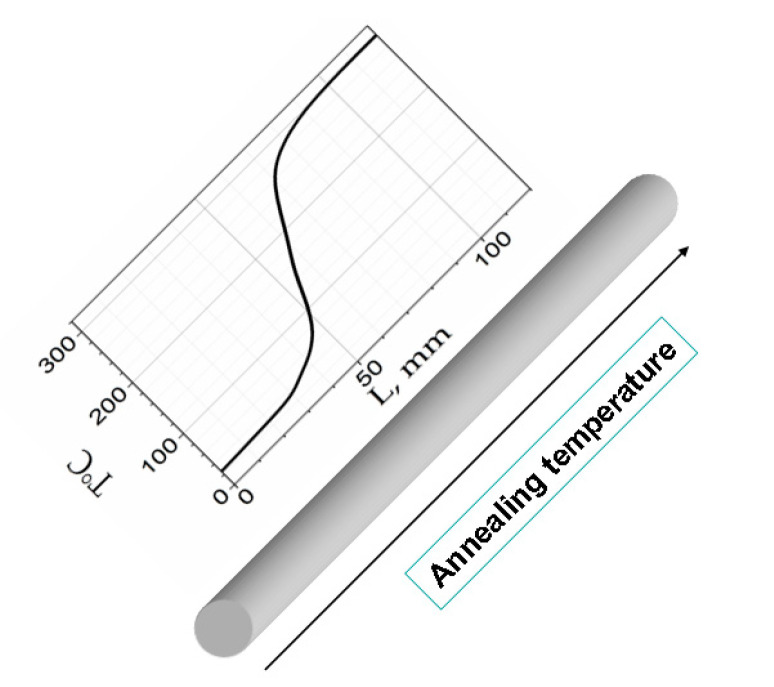
Distribution of annealing temperature *T_ann_* in sample.

**Figure 3 sensors-23-03079-f003:**
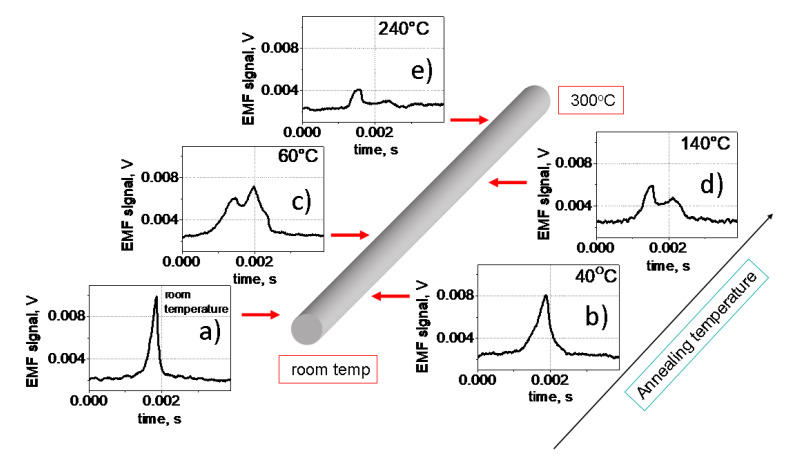
Distribution of EMF peak along the sample length. (**a**) room temperature, (**b**) 40 °C, (**c**) 60 °C, (**d**) 140 °C, (**e**) 240 °C.

**Figure 4 sensors-23-03079-f004:**
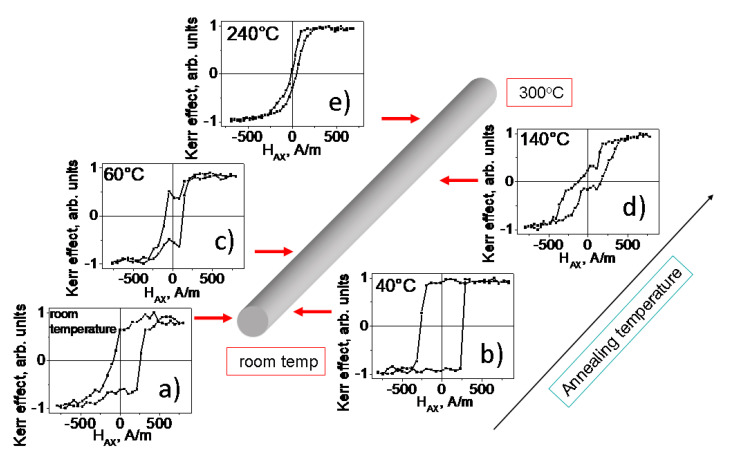
Distribution of MOKE hysteresis loops along the sample length. (**a**) room temperature, (**b**) 40 °C, (**c**) 60 °C, (**d**) 140 °C, (**e**) 240 °C.

**Figure 5 sensors-23-03079-f005:**
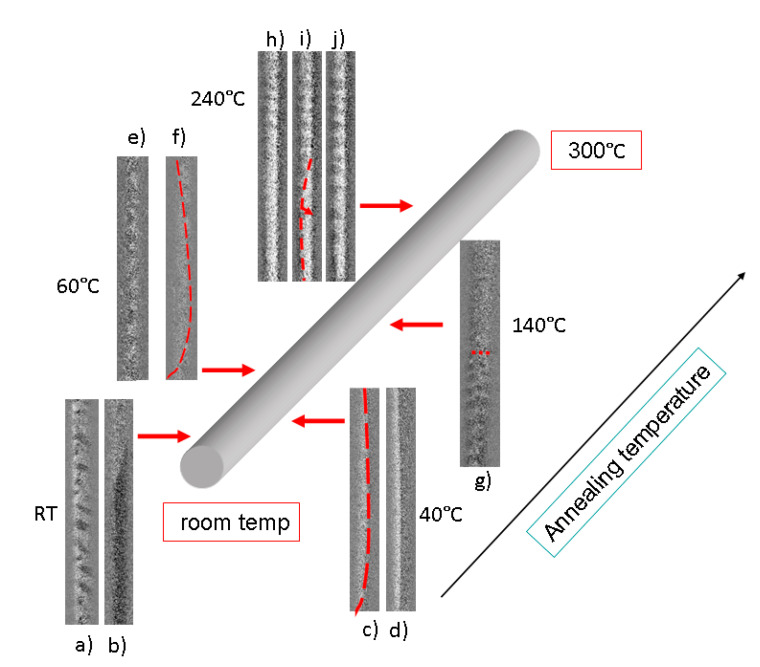
Distribution of surface domain structure along the sample length. (**a**) multi-domain structure, (**b**) inclined elliptic domain wall, (**c**) long curved domain wall, (**d**) longitudinally oriented domain wall, (**e**) multi-domain structure, (**f**) curved domain wall, (**g**) combination of two types of domain structure, (**h**–**j**) consistent motion of inclined spiral domains. Red dashed lines show the positions of the fronts separating the different types of domains.

**Figure 6 sensors-23-03079-f006:**
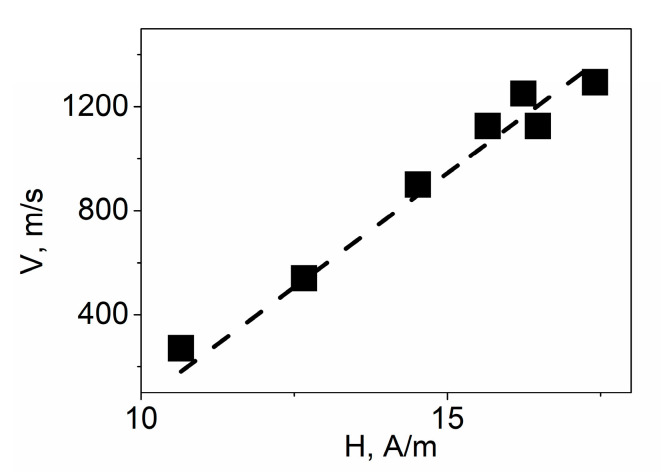
Dependence of the velocity of the DW on the external magnetic field in the area of the microwire corresponding the room temperature.

**Figure 7 sensors-23-03079-f007:**
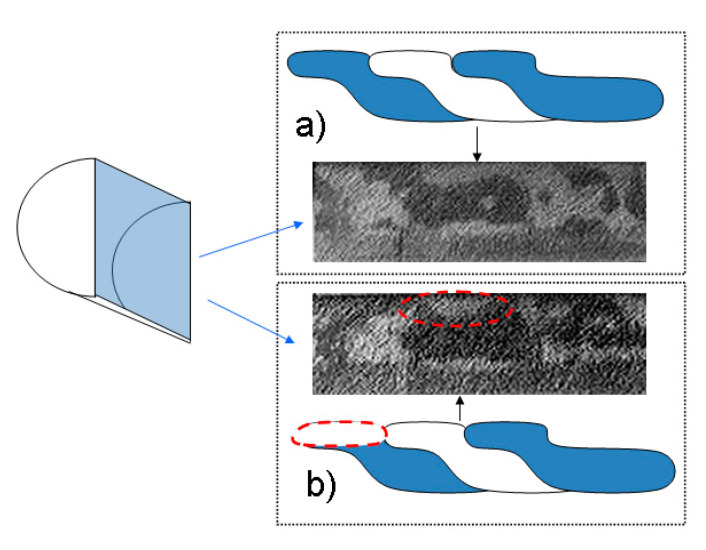
Domain structure in the surface of half-tube shaped microwire. (**a**) H_AX_ = 0.8 Oe, (**b**) H_AX_ = 0.4 Oe.

**Figure 8 sensors-23-03079-f008:**
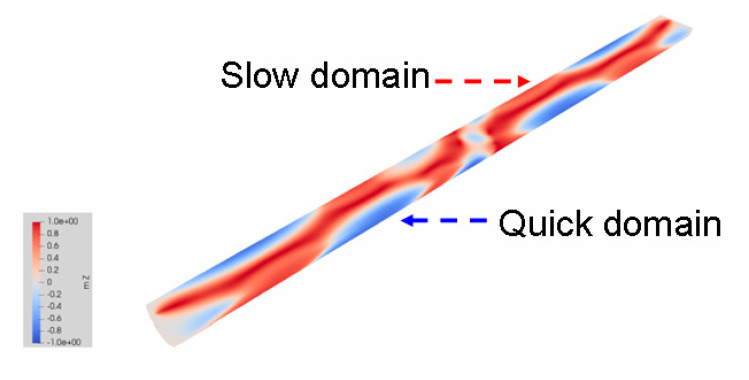
Calculated domain structure presented as half-tube.

**Figure 9 sensors-23-03079-f009:**
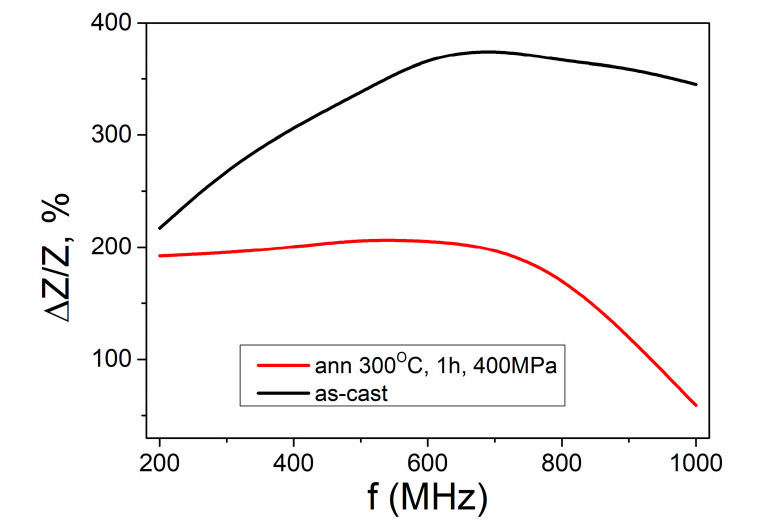
GMI dependences measured in the locations corresponding to room temperature and to annealing temperature of 300 °C.

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
