# Peer review of "Determination of Magnetic Structures in Magnetic Microwires with Longitudinally Distributed Magnetic Anisotropy"

_sensors, 2023, doi:10.3390/s23063079_

Round 1

Reviewer 1 Report

1) How the authors ensured temperature at different locations of a metallic wire. Normally one expect temperature should not be drastic different due to high conductivity

2) A thermal model to be included to substantiate the claimed temperature gradation

3) Objective of the paper is poorly narrated. The originality and novelty should be clearly spelt out in the introduction. Otherwise hit appears yet another manuscript on the topic as there are several published reports.

4)How the location specific GMI was measured? A detailed methodology is required to be included.

5) A theoretical explanation must be included to explain differential distribution of slow and quick domains across the length of the wire

Author Response

1) How the authors ensured temperature at different locations of a metallic wire. Normally one expect temperature should not be drastic different due to high conductivity

The temperature was measured using a commercial (NiCr-Ni KEYSIGHT Technologies) thermocouple inside the furnace, in the orifice (through which the microwire is introduced into the furnace) and near the furnace. As a result of the performed measurement the distribution of the annealing temperature was obtained (Fig. 2). Longitudinal homogeneity of the chemical composition and the geometric dimensions allow us to count on a smooth temperature change between room temperature and 300°C over a sufficiently long length of the wire under study. The additional validity of the obtained temperature dependence was confirmed by the fact that in the course of the study we found a large variety of magnetic structures that smoothly replaced one another.

2) A thermal model to be included to substantiate the claimed temperature gradation

The theoretical basis, on which our confidence in the validity of our temperature measurements and conclusions was based, is the work:

“Magnetic properties and magnetocaloric effect in Heusler-type glass-coated NiMnGa microwires” A. Zhukov, V. Rodionova, M. Ilyn, A.M. Aliev, R. Varga, S. Michalik, A. Aronin, G. Abrosimova, A. Kiselev, M. Ipatov, V. Zhukova, Journal of Alloys and Compounds 575 (2013) 73–79.

In it, the authors investigated the magneto-caloric effect in microwires. For us, there were several important factors. Firstly, direct measurements were carried out in this work using the modulated method allowing the detection of a change in the temperature with an accuracy of below 10-3 K. Secondly, the attention was paid to the effect of the glass coating of the microwire on temperature changes in microwires. It was shown that after glass removal, the temperature changes increase by only no more than 0.22 K. The results of solving the time-differential heat dissipation equation and temperature distribution inside the microwire are also important. The most essential result of the theoretical calculations could be considered the radial distribution of the temperature within the microwire at different time moments. The thermal processes within the metallic core occur very fast. Temperature equilibrium occurs within 100 μsec. This gives us additional reason to consider our temperature measurements as valid.

3) Objective of the paper is poorly narrated. The originality and novelty should be clearly spelt out in the introduction. Otherwise hit appears yet another manuscript on the topic as there are several published reports.

The originality and novelty of this research is initially determined by our original idea to create samples with a continuous and uniform distribution of magnetic properties. The longitudinal distribution of the annealing temperature is a natural reason for the distribution of magnetic anisotropy, which in turn leads to a continuous distribution of magnetic properties along the length of the microwire. It is the continuity of changes in properties that is the object of our close scientific interest. Having carried out the first studies of such samples, in particular, the influence of graded magnetic anisotropy on domain wall propagation, we discovered the lack of knowledge about the features of the magnetic structure in this type of samples. Although we previously studied the magnetic structure in various annealed samples, it is the continuity of the change in the magnetic properties of the microwire in a wide temperature range from room temperature to 300°C that is the decisive factor. Previously, we "locally" examined various annealed samples, and as it turned out, thereby missed and did not fix various essential properties. Now this shortcoming has been eliminated. In particular, for the first time we observe the effect of the coexistence of various magnetic structures, replacing each other both in the process of magnetization reversal and when moving along the sample.

4) How the location specific GMI was measured? A detailed methodology is required to be included.

The impedance dependencies were evaluated using the specially designed micro-strip sample holder. The holder was placed inside a sufficiently long solenoid that creates a homogeneous magnetic field. The microwire impedance was determined from the reflection coefficient measured by the vector network analyzer [13, 14]. We evaluated the magnetic field dependencies of the GMI ratio, ΔZ/Z, defined as:

ΔZ/Z = [Z(H) - Z(Hmax)]/Z(Hmax) (1)

where Hmax is the maximum applied magnetic field. Use of mentioned technique allows to measure GMI effect in the extended frequency range up to 1 GHz.

The GMI dependencies were measured for the locations in the studied microwire corresponding to the room temperature and to annealing temperature of 300º. The small samples for GMI measurements were cut off from two opposite ends of the long sample, where the regions of interest to us were located. Other parts of the sample were not destroyed during this manipulation.

5) A theoretical explanation must be included to explain differential distribution of slow and quick domains across the length of the wire

The main publication of a theoretical nature, the results of which we were able to use in our studies of the dynamics of a domain wall in microwires, was

  1. Kladivová, J. Ziman, “Properties of a domain wall in a bi-stable magnetic microwire”, J. Magn. Magn. Mater. 480 (2019) 193–198, https://doi.org/10.1016/j. jmmm.2019.02.058.

 The results presented there were based on a model, which includes standard damping mechanisms (eddy current damping and spin relaxation damping) and the influence of the domain wall length on its mobility. It is this article that helped us to explain the differences in the dynamic properties of elliptical and spiral domain walls

Chizhik A, Zhukov A, Corte-León P, Blanco J M, Gonzalez J, Gawronski P (2019), “Torsion induced acceleration of domain wall motion in magnetic microwires,” J. Magn. Magn. Mater., vol. 489, 165420, doi: 10.1016/j.jmmm.2019.165420.

As far as the present results are concerned, our consideration is based on the idea that the isolated compact structures have higher mobility than the branched magnetic structures. The isolated structures are located near the surface, while the long branched structures are located more within the microwire, resulting in their reduced mobility.

Reviewer 2 Report

Type of manuscript: Article
Title: Determination of magnetic structures in magnetic microwires with
longitudinally distributed magnetic anisotropy
Journal: Sensors -
2238799

Review

More than 20 years of research progress regarding the nondestructive testing method of metal magnetic memory is reviewed and summarized in detail. From analyzing the current state of research on the method of magnetic memory, some key problems and future developmental trends are proposed. Although the research on the magnetic memory method has made great progress, the practical application still faces problems such as complex influencing factors and less quantitative research. In the future, for the magnetic memory method, it is necessary to strengthen the microscopic observations of magnetic domains, experiments of magneto-mechanical constitutive, establishment of quantitative models, modeling of complex influencing factors, and the study of identification, inversion, and criteria.

The authors have an experience in the investigation of microwires and their magnetic properties. In this work, they show recent results of the study of magnetic structure and magnetization reversal processes in magnetic microwires with distributed magnetic anisotropy caused in turn by distributed annealing temperature. The work is well written and presented, all results were discussed and explained. All this facilitates the reading of the manuscript. The influence of the annealing process at variable temperature on the dynamics of the magnetic domains in these microwires is interesting and important for possible applications in magnetic sensors.

I think that the topic of this study is quite interesting and that the results are in principle suitable for publication in Sensors. However, I would like the authors to answer the question below before the publication of this manuscript can be recommended.

  1. What are the novelties and importance of this work compared to the study published in that same journal, whose DOI is:
    https://doi.org/10.3390/s20247203 ? The investigated material has the same properties in both studies.

Author Response

What are the novelties and importance of this work compared to the study published in that same journal, whose DOI is:
https://doi.org/10.3390/s20247203 ? The investigated material has the same properties in both studies.

The originality and novelty of this research is initially determined by our original idea to create samples with a continuous and uniform distribution of magnetic properties. The longitudinal distribution of the annealing temperature is a natural reason for the distribution of magnetic anisotropy, which in turn leads to a continuous distribution of magnetic properties along the length of the microwire. It is the continuity of changes in properties that is the object of our close scientific interest. Having carried out the first studies of such samples, in particular, the influence of graded magnetic anisotropy on domain wall propagation, we discovered the lack of knowledge about the features of the magnetic structure in this type of samples. Although we previously studied the magnetic structure in various annealed samples, it is the continuity of the change in the magnetic properties of the microwire in a wide temperature range from room temperature to 300°C that is the decisive factor. Previously, we "locally" examined various annealed samples, and as it turned out, thereby missed and did not fix various essential properties. Now this shortcoming has been eliminated. In particular, for the first time we observe the effect of the coexistence of various magnetic structures, replacing each other both in the process of magnetization reversal and when moving along the sample.

As for the article mentioned by the reviewer, the novelty of the present research in comparison with the mentioned article is as follows. First, we would like to note that the repeated testing of samples with the same chemical composition is a common scientific practice, which we also follow. It is this practice that allows one to penetrate deeper into the understanding of the characteristics of the substance under study, thereby ensuring the systematic research. In our particular case, as mentioned above, the continuity of the change in magnetic properties is important, which is determined by the wide range of annealing temperatures to which the sample was subjected. As a basis, we took a sample with a chemical composition, about which we had certain initial information. This gave us grounds to roughly predict what we later observed in the experiment. Having created an original sample with the same chemical composition, but with properties continuously changing along the sample length, we moved to a new level of our research. Also it should be noted that it is the magneto-optical techniques that we use, that allow us for the first time to study in detail, with sufficiently high accuracy, the changes of the magnetic structure that occur along an extended sample. 

Reviewer 3 Report

This work presents a study on determination of magnetic structures in magnetic microwires with longitudinally distributed magnetic anisotropy. The paper is acceptable for publication in the present form but some suggestions should be considered by the authors:

1) The authors must give some real images of the samples - for example SEM and TEM images. In this way the grain size would be estimate. Also the real picture of the microwires have to be shown.

2) The authors have to show the cross section of the samples to the core -shell  ratio at several position of the microwires. This information could be related with the magnetic properties of the microwires.

Author Response

1) The authors must give some real images of the samples - for example SEM and TEM images. In this way the grain size would be estimate. Also the real picture of the microwires have to be shown.

The crystallization and Curie temperatures of the studied microwire, evaluated by the differential scanning calorimetry (DSC) method as described elsewhere [15], were about 510°C and 370°C, respectively. The crystallization and Curie temperatures values are consistent with the values observed in CoFe microwires [16] and ribbons [17] with similar compositions. Consequently, like other Co-rich stress-annealed microwires, the studied microwires annealed at 300°C retain amorphous structure after stress-annealing [18]. The amorphous structure of stress-annealed microwires is also indirectly confirmed by the excellent magnetic softness. In the XRD pattern of the studied sample a wide halo typical for amorphous materials can be observed.

2) The authors have to show the cross section of the samples to the core -shell  ratio at several position of the microwires. This information could be related with the magnetic properties of the microwires.

The problem of core-shell ratio has been discussed in detail in the work [10]. In this work, it was shown that the distributed annealing could create a medium with an artificially changed core inner radius. Changing the volume of the internal axially magnetized core made it possible to modify the DW dynamics. In the present paper, we are considering a modified core-shell model that assumes a branched inner core in the microwire. Nevertheless, the previously discovered change in core-shell ratio [10] contributes to the results obtained.
